# Manganese Removal Using Functionalised Thiosalicylate-Based Ionic Liquid: Water Filtration System Application

**DOI:** 10.3390/molecules28155777

**Published:** 2023-07-31

**Authors:** Ain Aqilah Basirun, Wan Azlina Wan Ab Karim, Ng Cheah Wei, Jiquan Wu, Cecilia Devi Wilfred

**Affiliations:** 1Centre of Research in Ionic Liquids (CORIL), Institute of Contaminant Management (ICM), Universiti Teknologi PETRONAS, Seri Iskandar 32610, Perak, Malaysia; ainaqilahbasirun@gmail.com; 2Faculty of Engineering, Universiti Putra Malaysia (UPM), Serdang 43400, Selangor, Malaysia; wanazlina@upm.edu.my; 3Camfil Malaysia Sdn Bhd, Plot 9A & 9B, Lorong Bemban 1, Bemban Industrial Estate, Batu Gajah 31000, Perak, Malaysia; ng.cheahwei@camfil.com (N.C.W.); jiquan.wu@camfil.com (J.W.); 4Fundamental and Applied Sciences Department, Universiti Teknologi PETRONAS, Seri Iskandar 32610, Perak, Malaysia

**Keywords:** thiosalicylate ionic liquid, manganese, water filtration system, response surface methodology

## Abstract

Aiming at the generation of new functionalised thiosalicylate-based ionic liquids, a polymeric hydrogel consisting of 1-hexylimidazole propionitrile thiosalicylate [HIMP][TS], with a solid biomaterial support based on polyvinyl alcohol (PVA)–alginate beads, was produced. This study aimed to develop a treatment method for removing manganese (Mn) heavy metal from industrial wastewater, which is known to be toxic and harmful towards the environment and human health. The method utilised an adsorption-based approach with an alginate adsorbent that incorporated a functionalised thiosalicylate-based ionic liquid. The synthesised smooth round beads of PVA–alginate–[HIMP][TS] adsorbent were structurally characterised using Fourier transform infrared spectroscopy (FTIR) and field emission scanning electron microscopy (FESEM). The Mn concentration and removal efficiency were evaluated using atomic absorption spectroscopy (AAS). Three important parameters were evaluated: pH, adsorbent dosage, and contact time. During optimisation using the interactive factor design of experiments through the Box–Behnken model, the results showed that the system achieved a maximum Mn removal efficiency of 98.91% at an initial pH of 7.15, with a contact time of 60 min, using a bead dosage of 38.26 g/L. The beads were also tested in an available water filtration prototype system to illustrate their industrial application, and the performance showed a removal efficiency of 99.14% with 0 NTU total suspended solid (TSS) and 0.13 mg/L turbidity analysis. The recyclability of PVA–alginate–[HIMP][TS] beads using 0.5 M HCl resulted in four cycles with constant 99% Mn removal. The adsorption capacity of Mn was also determined in optimum conditions with 56 mg/g. Therefore, the alginate–thiosalicylate-based ionic liquid system is considered an effective and environmentally friendly method for removing Mn heavy metal due to the high removal efficiency achieved.

## 1. Introduction

Water resources are important for social and economic development, and the United Nations (UN) sustainable development goals (SDGs) describe clean water as a basic human and environmental need. Generally, acceptable conditions for the discharge of industrial effluent such as heavy metal constituents follow the Malaysian standard, which considers two acceptable conditions of 80 mg/L and 150 mg/L for the upstream and downstream of sensitive environmental resources, respectively.

Industrial water faces a dual problem of strong acidity and the presence of heavy metals [1]. Manganese pollution in a water stream is easily identified through colour, characteristic odour, and taste. The threshold limit of manganese in potable water is 0.05 mg/L, while it must fall within the range of 0.01–0.2 mg/L for industrial use water. Manganese deposition in pipes during the transportation of water causes harmful effects to humans. At an inhalation rate exceeding 10 mg per day, manganese is reported to cause damage to the nervous system, and most metals are reported to cause renal and liver disorders at high exposure levels [2,3,4].

Adsorption has been shown to have a simple design, high performance, ease of operation, and low cost. Adsorption materials are widely available (e.g., agricultural and industrial wastes), and the process is swift, taking place within a few minutes to hours [5]. Many sorption studies have been conducted using the biomass of animals, plants, and other byproducts on metals, dyes, and substances alike [6]. An effective biodegradable adsorbent for the extraction of manganese from the effluents meets the ‘green’ process requirements in industry. Adsorption using ionic liquid through liquid–liquid adsorption is currently receiving interest for its excellent metal extraction features due to its low volatility, chemical stability, and chelating abilities.

Numerous studies have been reported with regard to investigating the ability of ionic liquids in heavy metal removal. Ionic liquids are molten salts composed of organic cations and various anions. The property of an ionic liquid is dependent on its cation and anion constituents. Hence, altering the cations and anions of the ionic liquid allows for the fine-tuning of favourable properties for specific applications. The most common cations used in ionic liquids include pyridinium, imidazolium, and phosphonium ions, whereas the typical anions include Cl^−^, Br^−^, BF_4^−^_, PF_6^−^_, NTf_2^−^_, and CF_3_SO_3_. In the context of metal ion extraction, this tunability property allows for the ionic liquid to be functionalised for the extraction of metal ions. The use of a functionalised hydrophobic ionic liquid consisting of imidazolium and thiosalicylate ions is reportedly favourable for metal ion extraction [7]. The presence of desired functional groups in this ionic liquid can effectively interact with different metal ions, which secondarily functions to facilitate the transport of ions into the organic phase, thus resulting in a better separation efficiency [8,9].

To better reflect the application of ionic liquid for heavy metal extraction, a Scopus keyword search focusing on the topics of “ionic liquid”, “metal extraction”, and “thiosalicylate” yielded 15 journal articles from 2010 to 2022. The VOS Viewer software was used to create the keyword co-occurrences of the documents, as displayed in Figure 1. A bibliometric assessment of the research landscape of functionalised ionic liquid was conducted to determine the versatility of metal extraction [5]. The highest relatedness or the greatest number of occurrences in previous articles was related to a discussion on improving the practicality of ionic liquid by introducing an immobilizing agent to support the ionic liquid, such as polyvinyl alcohol (PVA) or sodium alginate [7,9,10]. The search also revealed that the use of thiosalicylate-based ionic liquid as an adsorbent has been scarcely investigated. Thus, this present study is the first to use a thiosalicylate-based ionic liquid encapsulated into a crosslinked PVA and alginate hydrogel as a solid medium to remove manganese from real industrial wastewater [11].

Figure 2 shows the top 10 subjects with the most publications in adsorption studies using ionic liquid. Chemistry and Environmental Science had the highest number of publications in the Scopus database with 24.2% of documents published in the last 10 years, and over 4000 articles were found on the web-search engine. Other areas like chemical engineering, engineering, materials science, and biochemistry also had significant numbers of records. The growing concern for water quality by environmentalists and the successful utilisation of ionic liquid in extracting the recalcitrant molecules from polluted water are driving the increase in novelty and findings when researching this topic, which is expected to continue in the future, with more patents being filed [12].

The modified sorbents such as solvent impregnated resins, microcapsules, chitosan beads, activated carbon, silica gel, and polymeric sorbents and the magnetic nanoparticles among the most sorbent materials have received much attention for various metal extraction methods [9]. Most of the methods are based on the modification of physisorption and chemisorption mechanisms. However, chemical modification techniques provide high stability and reusability. The immobilisation of ionic liquids helps to reduce the amount of used ILs and enhances the efficiency of adsorbent Ils or the Ils themselves as a metal extractant [13]. Alginate is a linear polysaccharide that contains β-d-mannuronate (M) and α-l-guluronate (G) and has a high affinity for divalent cations such as Pb(II), Cu(II), and Cd(II). Due to its desirable properties, such as biocompatibility and non-toxicity, calcium alginate fibre has many potential applications. However, to improve the physicochemical properties of alginate, the polymer blends method can be used. Polyvinyl alcohol (PVA) is a synthetic polymer that is nontoxic, has high strength, and is ideal for enzyme and cell immobilisation [14]. PVA–alginate blends are physically stronger and more durable than the alginate ones, although both gels possess water contents extremely higher than their polymer contents. This biopolymer material can form a stable gel in the presence of divalent cations and has also been incorporated with other material such as polyvinyl alcohol and ionic liquids in order to enhance their efficiency as metal extractants [11]. The gelling is crucial for an effective adsorption performance and the endurance of the bead in harsh wastewater conditions. A proper understanding of adsorption mechanism is very important to assess the adsorption efficiency of the proposed adsorbent. The linkage between PVA and sodium alginate is shown in Figure 3. These studies provide a solid foundation for further research into the linkage between PVA and sodium alginate in functionalizing Ils and demonstrate the potential for using this approach to create new, improved materials with enhanced functional properties [15].

## 2. Results

### 2.1. Characterisation of [HIMP][TS] and PVA–Alginate–[HIMP][TS]

The ^1^H-NMR spectra of the synthesised [HIMP][TS] ionic liquid is depicted in Figure 4. From the spectrum, the expected chemical shifts (d) and splitting patterns are identifiable, indicating that the ionic liquid obtained matched the connectivity of atoms and thus confirms the chemical structure of the [HIMP][TS] obtained. The chemical shifts recorded in the spectrum along with the assignments to the corresponding chemical structure of the ionic liquid are listed as follows. Chemical shifts between 7.1 and 7.9 ppm could clearly be assigned to the aromatic hydrogen atoms of the thiosalicylate anion. The ^1^H-NMR of thiosalicylate anion in [HIMP][TS] is summarised as: δ 7.22 (2H, CH2), δ 7.47 (1H, CH), and δ 7.84 (1H, CH). Meanwhile, the 1H-NMR of 1-hexylimidazole propionitrile cation in [HIMP][TS] is summarised as: δ 0.83 (2H, CH2), δ 1.23 (6H, CH2), δ 1.69 (2H, CH), δ 3.89 (1H, CH), δ 3.22 (1H, CH), δ 3.92 (2H, CH2), δ 7.48 (2H, CH), and δ 7.9 (1H, CH). The chemical shift for the NMR solvent, deuterated dimethyl sulfoxide (DMSO), is 2.5 ppm.

FTIR qualitatively observed the presence of functional groups on the molecules. Figure 5 shows the IR adsorption for [HIMP][TS] ionic liquid (blue), PVA–alginate bead (green), and PVA–alginate–[HIMP][TS] bead (brown). Typical peaks of aliphatic C–H stretching were detected in the 2800–2900 cm^−1^ region, which originated from the alkyl chain of the ionic liquid. The peak observed at around 1636 cm^−1^ corresponds to the conjugated C=O functionality present in the anion part of the [HIMP][TS] ionic liquid, overlapping with the carboxylate group in PVA–alginate linkage, as mentioned by Silva et al. (2023) [16]. The peaks at approximately 1620 cm^−1^ and 1370 cm^−1^ suggest the presence of C=C and CS functional groups in the aromatic and imidazole structures of the ionic liquid, respectively. Additionally, the region around 1440 cm^−1^ and between 1000 and 600 cm^−1^ were assigned to the bending and stretching vibrations of the aromatic C–H group.

The FTIR analysis of [HIMP][TS] ionic liquid showed absorption bands at 2958 cm^−1^ corresponding to -CH_2_ stretching, 1628 cm^−1^ for C–N stretching, and a small peak protruded at 2232 cm^−1^ for the S–H group. This is in accordance with the synthesis of thiosalicylate ionic liquid from a previous study [17]. The O–H vibration in PVA–alginate–[HIMP][TS] decreased, indicating an interaction with the imidazolium cation. Based on the analysis of the FTIR peaks found in the spectrum, it can be inferred that the synthesised [HIMP][TS] ionic liquid contains the characteristic functional groups that correspond to both the imidazolium cation and the thiosalicylate anion.

Figure 6a depicts the beads formation from PVA–alginate and [HIMP][TS] after air-dried for 24 h. The beads showed no shrinkage upon being air-dried. The estimated diameter was measured to be approximately 4 mm. The ratio of the PVA/alginate mixture improved the beads to become pearl-like, with a shiny smooth surface. SEM micrograph in Figure 6c shows that the alginate/PVA beads with a surface roughness of various dimensions indicated a larger surface area, as reported from a previous study by Rahman et al. (2021) [18]. Meanwhile, SEM micrograph in Figure 6d shows the development of a hollow porous area on the bead cross-section surface with the addition of 7% ionic liquid [HIMP][TS]. The presence of different pore sizes in the bead is favourable for the adsorption of Mn heavy metals as these pores can facilitate the accumulation of the ions extracted from wastewater samples. The porosity of the PVA–alginate–[HIMP][TS] was studied through a surface area analysis using BET surface area study. The morphology analysis was also coupled with EDX elemental investigation to determine the immobilisation of [HIMP][TS] ionic liquid onto the PVA–alginate beads. Figure 6f,g indicate the EDX spectra of the PVA–alginate–[HIMP][TS] bead after the Mn removal. The Mn element peak appears at several surface spectra of the beads, proving the attachment of Mn onto the beads.

### 2.2. Study of Adsorption Performance

#### 2.2.1. Effect of pH

The impact of wastewater pH on Mn removal through adsorption was examined. Three different pH levels of 3, 7, and 11 were tested to assess removal in acidic, neutral, and alkaline conditions, respectively. The results, shown in Figure 7, indicate that the highest Mn removal efficiency was achieved at pH 7, followed by pH 11 and pH 3. This can be explained by the fact that at a low pH (pH 3) of the wastewater or solution, an excess concentration of hydroxonium ions can cause metal cations to compete for binding sites on the adsorbent, thus reducing metal adsorption. Conversely, a high pH (pH 11) leads to an excess of hydroxide ions, causing the formation of metal hydroxide precipitates that limit adsorption and decrease metal removal efficiency. Based on these findings, pH 7 was determined to be the optimal pH for Mn removal from the wastewater studied. This conclusion aligns with the findings reported in a review by Rudi et al. (2020) [19], which stated that, when using adsorbent materials such as alginate for Mn removal, the optimal pH range is 5 to 7.

#### 2.2.2. Effect of Adsorbent Dosage

Figure 8 illustrates the effect of bead dosage on Mn extraction efficiency. It is observed that the removal percentages of Mn ions show an insignificant increment upon the increase in bead dosage of 10, 30, 50, and 100 g/L after 60 min of contact time. The selection of bead dosage is according to previous studies and concern the scale-up application of a filtration system prototype. Generally, increasing the adsorbent dosage at a constant Mn concentration provides more available adsorption sites of the beads, thus increasing the of Mn removal [20]. In contrast, the removal percentages of Mn ions for PVA–alginate–[HIMP][TS] beads were declined on the usage of an adsorbent dosage of more than 50 g/L. The maximum adsorptive removal was found for 50 g/L g of bead dosage at 96%. The decrease may be attributed to the saturation of active sites of the beads with Mn.

#### 2.2.3. Effect of Contact Time

The adsorption of Mn ions onto PVA–alginate–[HIMP][TS] beads was studied with contact times ranging from 10 to 120 min. Figure 9 reveals that the adsorption process was rapid during the first 5 min and reached equilibrium after 60 min. The results of the contact time study showed that the adsorption amount increased with the increase in adsorbate concentration. During the initial 5 min, the adsorption took place quickly as there were many empty sites on the surface of the adsorbent that could be occupied by Mn molecules. As the incubation period extended, the adsorption gradually declined as the sites on the adsorbent became saturated and the system reached equilibrium. The equilibrium time at various adsorbate concentrations increased with an increase in the reaction temperature. Each adsorbate concentration reached its equilibrium state at a different incubation period. Of all factors studied, contact time had the greatest impact on the adsorption performance, but a balance between an economically feasible incubation period and the percentage of sorption must be achieved for commercial purposes [21].

#### 2.2.4. Optimisation Study Using Surface Response Methodology

In this study, Box–Behnken design (BBD) was selected to study the relationship between factors. Experimental design and results (experimental and predicted value) of Box–Behnken for Mn removal by PVA–alginate–[HIMP][TS] are presented in Table 1. The main factors affecting adsorption process pH, adsorbent dosage, and contact time. Table 1 shows the actual and predicted responses for the removal percentage of Mn from the batch adsorption system. From the table, the highest removal percentage of Mn performed at run no. 21 (99.81%). ANOVA analysis (Table 2) validated the BBD model’s adequacy for experimentation, indicating that the model with an F-value of 54.05 and *p*-value of 0.0001 was statistically significant. The non-significant lack of fit (0.0708) fits the model well in comparison to the pure error, indicating randomness. The computed experimental R^2^ was 0.9819, which has a strong correlation with both the adjusted R^2^ (0.9636) and the expected R^2^ (0.9022).

Most of the empirically predicted values are in good agreement with the experimentally recorded data, including the replicated data. The signal-to-noise ratio is measured using “Adeq Precision”. A ratio greater than 4 is preferred. The ratio of 30.56 suggests that the signal is adequate. The coefficient of variation % (CV%), which can be used to assess the experiment’s reliability and precision, was found to be 7.003, which is significant. The higher the CV% value, the less trustworthy the experiment. The CV value is a measure of the data’s residual variance in relation to the size of the mean. A higher CV value shows that the standard deviation is relatively large compared to the mean. A lower CV value below 20 is considered acceptable [22]. The predicted residual sum of squares (PRESS) value was 39.55. It is a measurement of how well each point in the design fits together. The lower the PRESS value, the more accurately the model matches the data points. Prediction residual sum of squares (PRESS) is a form of cross-validation used in regression analysis to provide a summary measure of the fit of the model to a sample of observations that were not themselves used to estimate the model. It is calculated as the sum of squares of the prediction residuals for these observations. The outcome was subjected to multiple regression analysis, with the resulting second-order polynomial of Equation (1) in terms of the coded level reflecting Mn extraction (Y), respectively.
Y = +99.69 + 1.80A − 0.057B − 0.27C − 1.03D − 4.89A^2^ − 0.52B^2^ − 5.16C^2^ − 0.41D^2^ + 1.91AB + 2.74AC + 0.81AD + 0.86BC + 0.91BD − 0.10CD (1)

In comparison, results from OFAT and RSM were gathered and compared as presented in Table 3. A better extraction (%) was achieved through RSM optimisation.

The 3D response surface model graph and contour were constructed by using Design-Expert software. Each graph demonstrates the impact of mutual interaction between independent parameters tested. The contour plot corresponds to their effects on the dependent parameter, which, in this case, is the extraction efficiency. The purpose of response surface analysis is to determine the efficiency of the optimal condition variables to achieve a maximum or desirable response. An elliptical contour is found when there is a great interaction between the independent parameters [23]. Figure 10 shows the 3D response surfaces and contour plots of the model of interactions between the significant independent variables, (A) pH, (B) adsorbent dosage, (C) contact time, and (D) alum concentration towards the dependent variable, and the removal percentage of Mn from experimental design.

#### 2.2.5. Pseudo-First- and Pseudo-Second-Order Kinetic Analyses of Mn Adsorption Study Using PVA–Alginate–[HIMP][TS]

Figure 11 and Table 4 summarise the results of a study that compared the performance of two kinetic models, the pseudo-first-order and pseudo-second-order models, and the error function analysis of Mn removal using PVA–alginate–[HIMP][TS], respectively. The pseudo-second-order model exhibited a higher qt value (63.61 mg/g) than the pseudo-first-order model (54.41 mg/g), indicating that it provided a better fit to the data. Additionally, the RMSE value for the pseudo-second-order model (1.9) was lower than that of the pseudo-first-order model (2.3), indicating that the former was more accurate in predicting the data. Both models had high R^2^ and adjusted R^2^ values, indicating that they fit the data well without overfitting. The AICc and HQC values were lower for the pseudo-second-order model, indicating that it was a better fit to the data. However, the higher qt value and lower RMSE value of the pseudo-second-order model indicated that it was a more appropriate model for predicting Mn adsorption using PVA–alginate–[HIMP][TS]. These results could have important implications for the development of more effective and efficient adsorption systems for removing Mn from aqueous solutions. Previous studies by Li et al. [24], Mohan et al. [25], and Xu et al. [26] have found that the pseudo-second-order model provides a better fit to the data with a higher maximum adsorption capacity compared to the pseudo-first-order model for Mn adsorption from aqueous solutions using different adsorbents, in agreement with the present data.

### 2.3. Application of Mn Adsorption Using Water Filtration System

The feasibility of using PVA–alginate–[HIMP][TS] beads adsorbent for industrial purposes was evaluated through a water filtration prototype system, with Table 5 summarizing the testing conditions utilised in the prototype system and the performance of the Mn removal. It was expected that a longer contact time would be necessary to observe the overall adsorption trend of the beads at the specified dosage due to the higher volume of wastewater used. A total of 120 min of the treatment process resulted in 99.14% of Mn removal from 10,000 ppm of concentrated Mn wastewater. The impurities in the sample were sedimented using 5 g/L of alum prior the adsorption process. As it passed through the cellulose filtration, the sample was collected. Turbidity and the total suspended solid were almost cleared after the filtration process.

### 2.4. Reusability Study

The recyclability study of PVA–alginate–[HIMP][TS] beads for removing 10,000 ppm of Mn from industrial wastewater indicates that the beads can be used for up to four cycles. Based on Figure 12, the initial Mn removal efficiency was high (98.62% to 98.93%), but it decreased significantly in the later cycles (92.12% to 93.35%), suggesting a loss of adsorption capacity over time. To improve recyclability, the beads can be regenerated using appropriate desorbing agents such as dilute acids or bases. This can restore the adsorption capacity of the beads and enable their use for subsequent cycles of Mn removal. Previous studies have investigated the recyclability of adsorbents to remove Mn from various sources of wastewater. For example, one study focused on the use of activated carbon for removing Mn from drinking water and found that the activated carbon could be reused for up to four cycles with an average Mn removal efficiency of 92.7% [27]. Another study evaluated the use of Fe-based magnetic nanoparticles for removing Mn from groundwater and demonstrated that the nanoparticles could be reused for up to six cycles with an average Mn removal efficiency of 87.3% [28]. In addition to the recyclability study, it is also important to determine the reliable concentration range of Mn that can be effectively removed by the PVA–alginate–[HIMP][TS] beads. This can be performed by conducting a batch adsorption study at different initial Mn concentrations, particularly on real wastewater content, and analysing the Mn removal efficiency. The results can be used to determine the maximum Mn concentration that can be reliably removed by the beads.

### 2.5. The significance of Hexylimidazolium Propionitrile Thiosalicylate Ionic Liquid towards Adsorption

The use of ‘ionic liquid’ encapsulated in a solid support such as PVA–alginate hydrogel results in a more effective adsorption compared to without using an ionic liquid. Previous studies have reported that the application of PVA–alginate hydrogel without an ionic liquid resulted in a 50% Mn adsorption efficiency [11], while the addition of chloride ionic liquid improved the efficiency to 90% [18]. The enhancement observed in this recent study demonstrates nearly a 100% increase in selectivity for adsorption when thiosalicylate-based ionic liquid is used. This is because ionic liquids are a diverse class of materials known for their unique properties and characteristics. These properties can vary depending on the specific composition of the ionic liquid, including the choice of cations and anions [29]. Ionic liquids generally exhibit low volatility, a wide liquid temperature range, high thermal stability, non-flammability, good solvating power, and the ability to be tailored for specific applications [30].

Encapsulating thiosalicylate-based ionic liquid in PVA–alginate hydrogel beads provides several advantages as a Mn adsorbent. It enhances adsorption capacity through the strong affinity of thiosalicylate ligands for Mn, ensuring an improved efficiency. The adsorbent demonstrates selectivity for Mn ions while minimizing interference from other metals. Encapsulation within the hydrogel matrix maintains stability, preventing leaching and degradation. The bead form enables easy handling and separation. The adsorbent can be regenerated and reused, reducing waste and costs. Additionally, the use of biocompatible, biodegradable materials aligns with environmentally friendly practices. Overall, this encapsulated ionic liquid system combines an enhanced capacity, selectivity, stability, ease of use, reusability, and eco-friendliness, offering promise for efficient Mn removal from wastewater [31].

### 2.6. Adsorption Mechanism

Understanding the mechanism of adsorption is crucial to evaluate the efficiency of an adsorbent. The adsorption of Mn with encapsulated PVA–alginate–[HIMP][TS] involves a charge-induced mechanism, which is characterised by synergistic effects of π–π interactions, electrostatic interactions, and hydrophobic interactions. The positive N^+^ atom of the imidazolium ring and negative thiosalicylate anion of [HIMP][TS] enable adsorbates such metal ions to interact electrostatically. Aromatic moieties in adsorbent and adsorbates can lead to an enhanced adsorption by interactions with the metal ions [32].

## 3. Material and Methods

### 3.1. Synthesis and Characterisation of [HIMP][TS] and PVA–Alginate–[HIMP][TS]

The ILs used in the present study were synthesised using chemicals of analytical grade. The CAS number, source, and grades of the chemicals used are as follows: imidazole (288-32-4, Sigma-Aldrich (St. Louis, MO, USA) 99%), acetone (67-64-1, Sigma-Aldrich 99.8%), acetonitrile (107-13-1, Aldrich 99%), anhydrous methanol (67-56-1, Sigma-Aldrich 99.8%), 1-chlorohexane (111-15-1, Aldrich 99%), sodium hydroxide pellets (Merck (Darmstadt, Germany), 99%), thiosalicylic acid (Sigma-Aldrich, 97%), polyvinyl alcohol (Sigma-Aldrich, MW = 30,000), sodium alginate (R&M Chemicals (Chandigarh, India), >95%), acetone (Merck, >95%), calcium chloride (R&M Chemicals, 99.5%), boric acid (Merck, >95%), hydrochloric acid (R&M Chemicals, 37% fuming), and 3-chloropropionitrile and diethyl ether (60-29-7, Sigma-Aldrich 99%).

### 3.2. Preparation of Real Industrial Manganese (Mn) Wastewater

Real wastewater containing approximately 100,000 ppm of manganese was collected from an air filter company (Camfil Sdn Bhd) near Batu Gajah, Perak. The actual concentrated wastewater was diluted into 1000 dilution factor for analysis. For the application on the filtration system, the actual concentrated Mn wastewater was utilised.

### 3.3. Synthesis of Hexylimidazolium Propionitrile Thiosalicylate [HIMP][TS] Ionic Liquid

Synthesis of [HIMP][TS] included three consecutive reactions. All three pathways were followed the stoichiometry (1:1) ratio. The method was adapted from and followed that of Rahman et al. (2021) [18]. First, 0.012 mole of imidazole and potassium hydroxide was crushed and dissolved in 100 mL of dimethyl sulfoxide. 1-chlorohexane was added dropwise while stirring the mixture in an ice bath to prevent exothermic reaction. The mixture was left to stir for 24 h at 25 °C for a complete reaction. Distilled water was added into the mixture with the ratio 3:1 and to chloroform (1:1) to wash the unreacted reactant, and the sample was collected from the bottom through liquid–liquid extraction and a rotary evaporator. The second pathway involves the mixture of 0.05 mole of synthesised 1-hexylimidazole with 3-chloropropionitrile. The mixture was stirred and heated at 55 °C for 48 h. The sample was then washed by using diethyl ether as a solvent and the ionic liquid [HIMP][Cl] was collected through rotary evaporator. The final pathway of the synthesis, including the metathesis of anion, exchanged between chloride and thiosalicylate as anion. An amount of 0.06 mole of thiosalicylic acid and sodium hydroxide were dissolved in 100 mL of methanol as solvent. The mixture was allowed to dissolve completely and [HIMP][Cl] was added into the mixture. The reaction was stirred at 500 rpm at 25° for 24 h for complete reaction and the [HIMP][TS] was obtained with NaCl as byproduct. NaCl was the filtered using acetonitrile as a solvent and through rotary evaporator. The clear, viscous yellow of newly synthesised [HIMP][TS] was analysed and characterised using ^1^H NMR and FTIR analyses and the reaction pathways are summarised in Figure 13.

### 3.4. PVA–Alginate Ionic Liquid Hydrogel Beads Fabrication

An amount of 8 g of PVA and 3 g of sodium alginate were dissolved in 100 mL of Millipore water and stirred at 80 °C for 8 h. [HIMP][TS] (7% *w*/*v*) was added to this mixture. The mixture was stirred at 30 °C for 6 h at a stirring rate of 500 rpm to obtain a homogeneous gel blend, which was then extruded into a gently stirred saturated 5% *w*/*v* CaCl_2_–boric acid solution using a syringe, which resulted in spherical hydrogel beads. The manual drip rate was approximately 0.01 mL or 1 drop every 4.5 s. After 24 h, the beads were washed with Millipore water to remove any impurities. The beads were air-dried for 24 h to prevent over-shrinking the beads. The bead formation is summarised in Figure 14. The surface characteristics of the beads were analysed using FESEM and FTIR analyses. The surface characteristics of the beads were analysed using scanning electron microscopy (SEM) after curing.

### 3.5. Mn Adsorption Optimisation Study

Efficiency of PVA–alginate–[HIMP][TS] beads adsorbent to remove Mn was evaluated in a batch adsorption study by varying three parameters: initial pH of the wastewater, adsorbent dosage, and contact time. Wastewater samples were prepared at 1000 dilution factor and Mn concentrations were measured using AAS. Removal efficiency was calculated using the following formula [33]:(2)$$Extraction\;efficiency(\%)=\frac{\left(C_{o}-C_{e}\right)}{C_{o}}\times 100\%$$
where C_o_ and C*_e_* (mg/L) are the liquid-phase concentrations of initial adsorbate and equilibrium, respectively.

The optimisation was performed using two methods of OFAT and RSM. The factors affecting the extraction efficiency, which include pH, PVA–alginate–[HIMP][TS] bead dosage, and contact time, were studied in batch adsorption scale study. The parametric experiments for the removal of manganese from aqueous solution were conducted under controlled agitation conditions. The influence of initial pH was studied in the range of 3.0–11.0 at a fixed manganese concentration of 10,000 mg/L, sorbent dosage of 10, 30, 50, and 100 g/L, and contact time 30–120 min. All the experiments were conducted at controlled conditions of ambient temperature and agitation speed of 150 rpm.

Box and Behnken (BB) created a 3-level incomplete factorial design as an alternative to the labour-intensive full factorial design [34]. Second-order polynomials must be employed in the modelling to accurately reflect linear, quadratic, and interaction effects. Box and Behnken devised this implementable design to reduce the quantity of experiments required, particularly in quadratic model fitting. Design-Expert software was used to analyse the response surface methodology (version 11, Stat-Ease Inc., Minneapolis, MN, USA). The importance of the polynomial’s complete terms is statistically analysed, yielding a probability value (*p* < 0.05). The mean of biosorption was utilised as the response in all tests, which were conducted in triplicate. The model based on multiple quadratic regressions was analysed using a second-degree polynomial model. The Box–Behnken experimental design for Mn adsorption is presented in Table 6.

### 3.6. Adsorption Kinetic Study

This approach is similar to the one used in batch equilibrium investigations. The concentration of the solution of the absorbent–absorbate solution at various time intervals was determined. The quantity of adsorbed at time t, denoted by the symbol qt (mg/g), was estimated using Equation (3). qt denotes the adsorption capacity by 1 g of adsorbent at a certain time of exposure.

Equation for pseudo–first-order kinetic model

The pseudo-first-order kinetic model equation is as follows [35]:(3)$$q_{t}=q_{e}\left(1-e^{-k_{1}t}\right)$$
Equation for pseudo-second-order kinetic model

The pseudo-second-order equation is as follows [36]:

The model is based on the adsorption capacity onto a solid phase and the nonlinear form of PSO was initially proposed by Blanchard et al. [37]. It is expressed as:(4)$$q_{t}=\frac{k_{t}qe^{2}t}{1+k_{2\;}q_{e}t}$$

The slope and intercept of the plot of t/qt vs. t result in the values for qe and k_2_, respectively.

### 3.7. Statistical and Error Function Analysis

#### 3.7.1. Fitting of the Data

Fitting of the adsorption kinetics and isotherms nonlinear data was carried out by nonlinear regression utilising the Marquardt algorithm, CurveExpert Professional software, Version 2.6 [33,38].

#### 3.7.2. Statistical and Error Function Analysis

A one-way ANOVA (with 95% confidence interval) was performed to examine the difference between parameters, with *p* < 0.05 being statistically significant.

For kinetics model determination, several statistical discriminatory methods, including Bayesian information criterion (BIC), AICc (Akaike information criterion), bias factor (BF), root-mean-squared error (RMSE), accuracy factor (AF), and adjusted determination coefficient (R^2^) were utilised [39,40].

### 3.8. Application of Mn Adsorption Using Water Filtration System

The aim of the project was to assess the efficacy of adsorbent PVA–alginate–[HIMP][TS] beads in removing Mn heavy metal from a water filtration prototype system (Figure 15). The prototype system composed of different water pre-treatment units, including mixing, coagulation, sediment, and pre-filter units, as well as an adsorption column unit where the adsorbent PVA–alginate–[HIMP][TS] beads was placed for testing. Optimised factors were carried into the application on the prototype. A volume of 200 mL of concentrated wastewater with a pH of 7 was filled into the filtration column unit via a tap valve, allowing it to contact 30 g of PVA–alginate–[HIMP][TS] beads for Mn adsorption for 60 min under continuous stirring. The process was prolonged until 120 min to determine the equilibrium adsorption process. The resulting treated wastewater samples were collected every 5 min for analysis, and the concentration of Mn before and after the adsorption process was assessed using AAS to determine the EE%.

## 4. Conclusions

In conclusion, the study developed a new adsorption-based approach using PVA–alginate–[HIMP][TS] beads for the efficient removal of Mn heavy metal from industrial wastewater. The alginate beads incorporated a functionalised thiosalicylate-based ionic liquid, which was structurally characterised and evaluated for Mn removal efficiency. The study optimised three important parameters of pH, adsorbent dosage, and contact time using the Box–Behnken design of experiment, achieving a maximum Mn removal efficiency of 98.91%. Moreover, the PVA–alginate–[HIMP][TS] beads were found to be recyclable for up to four cycles, with a 99% constant Mn removal rate, using 0.5 M HCl. The study highlights the potential of the alginate–thiosalicylate-based ionic liquid system as an effective and environmentally friendly method for Mn removal from industrial wastewater.

## Figures and Tables

**Figure 1 molecules-28-05777-f001:**
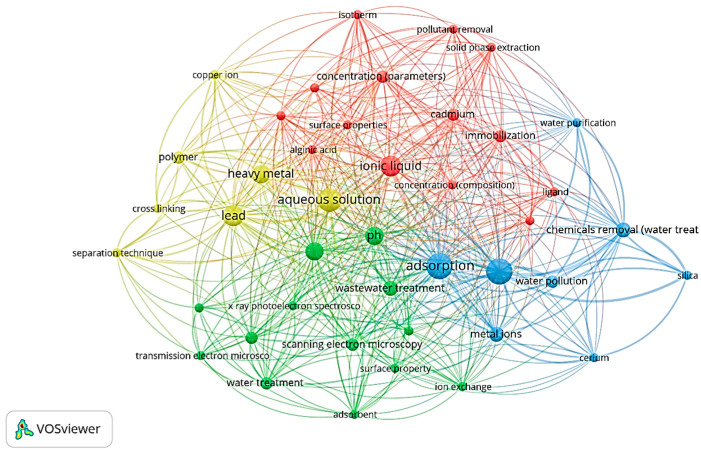
SCOPUS 2008–2023 keyword mapping based on the co-occurrence of author keywords in documents published in SCOPUS (keywords: “ionic liquid”; “metal extraction”; and “thiosalicylate).

**Figure 2 molecules-28-05777-f002:**
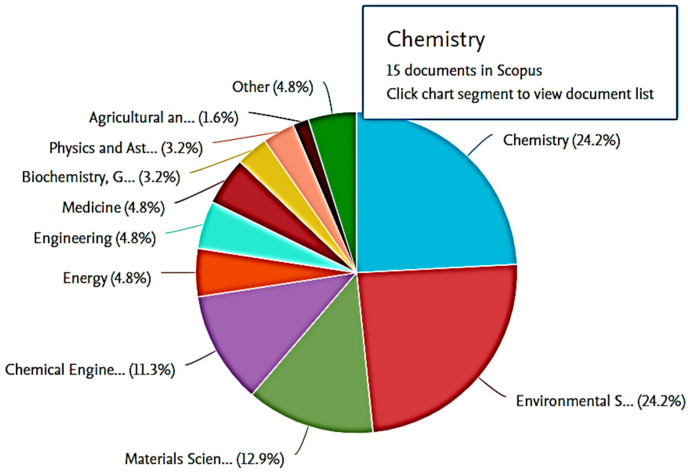
Cluster field study of co-occurrence research on functionalised ionic liquid from Scopus database from 2010 to 2022.

**Figure 3 molecules-28-05777-f003:**
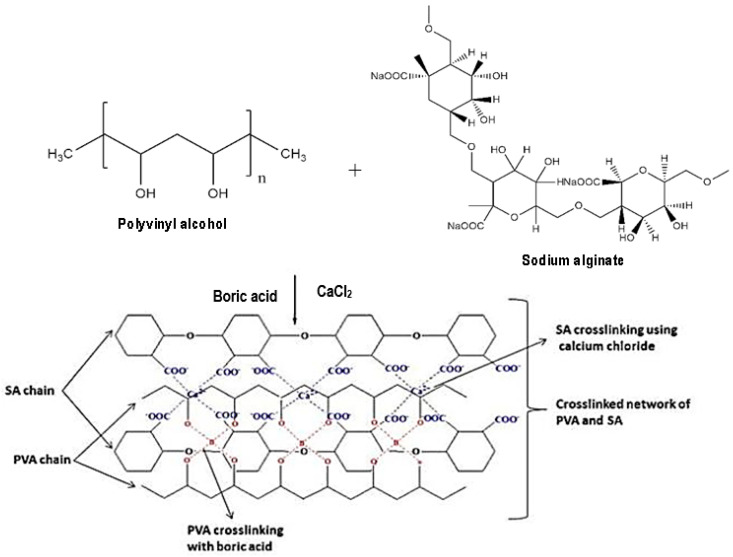
Schematic illustration of polyvinyl alcohol (PVA) and sodium alginate to form supramolecular structure of a PVA–alginate hydrogel (adapted from Nataraj and Reddy [15]).

**Figure 4 molecules-28-05777-f004:**
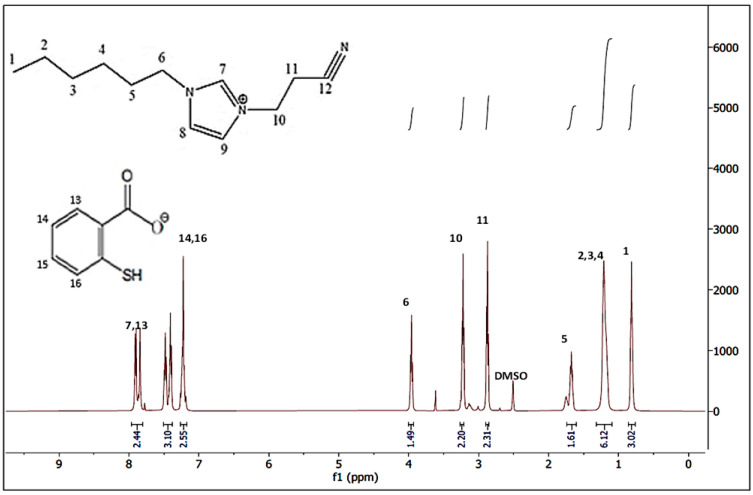
NMR spectrum of [HIMP][TS] ionic liquid.

**Figure 5 molecules-28-05777-f005:**
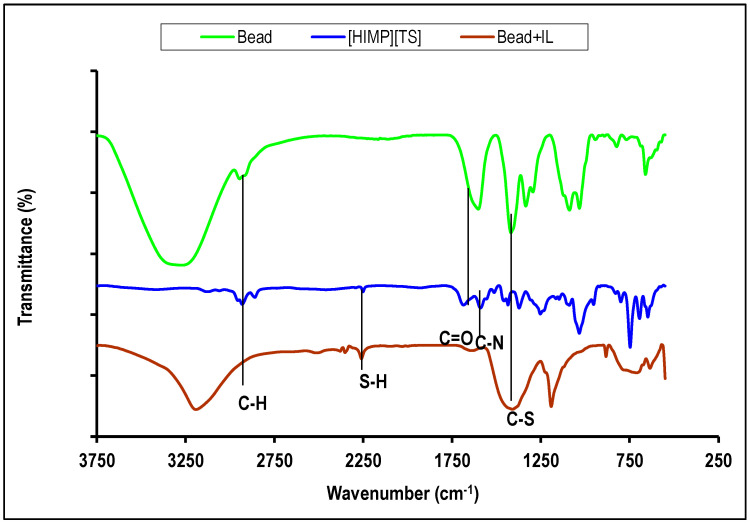
FTIR spectrum of [HIMP][TS] ionic liquid (blue), PVA–alginate bead (green), and PVA–alginate–[HIMP][TS] bead (brown).

**Figure 6 molecules-28-05777-f006:**
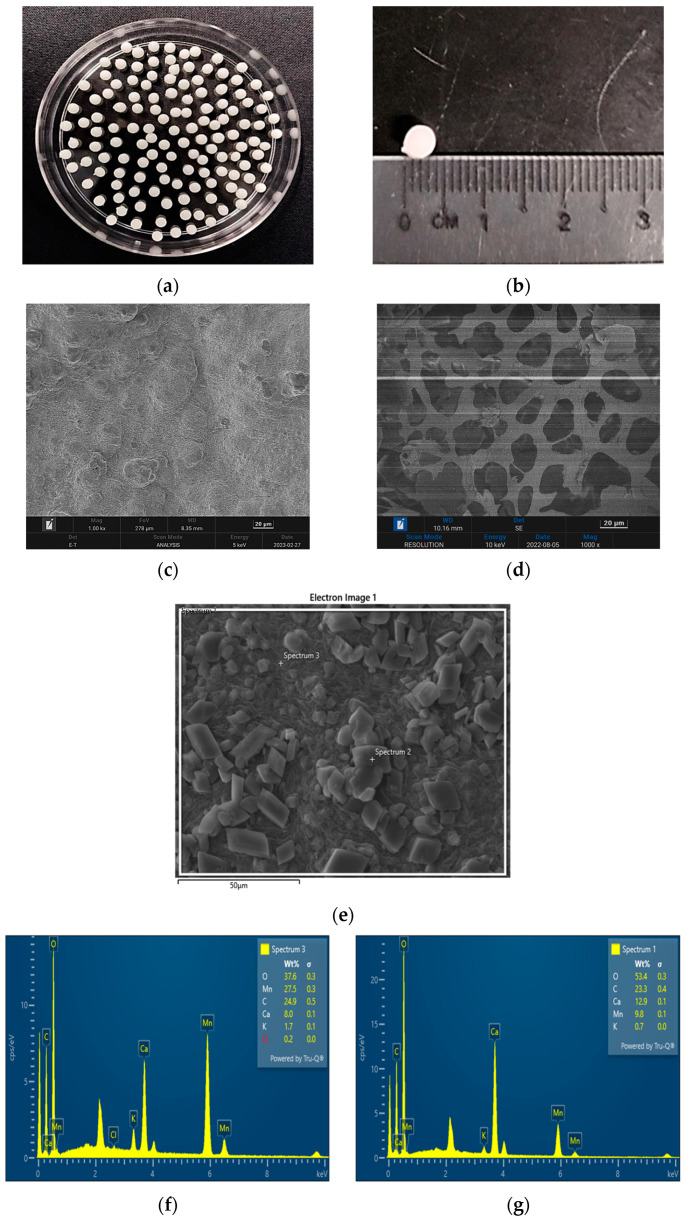
Bead formation form PVA–alginate and [HIMP][TS] (**a**). Bead size after 24 h air-dry: approximately 0.4 cm (**b**). SEM micrograph of PVA–alginate bead (magnification: 1000×) (**c**). SEM micrograph of PVA–alginate–[HIMP][TS] bead (magnification: 1000×) (**d**). Figure for EDX spectra analysis (**e**). EDX spectra of PVA–alginate–[HIMP][TS] bead after the Mn removal from spectra 1 and 2 (**f**,**g**).

**Figure 7 molecules-28-05777-f007:**
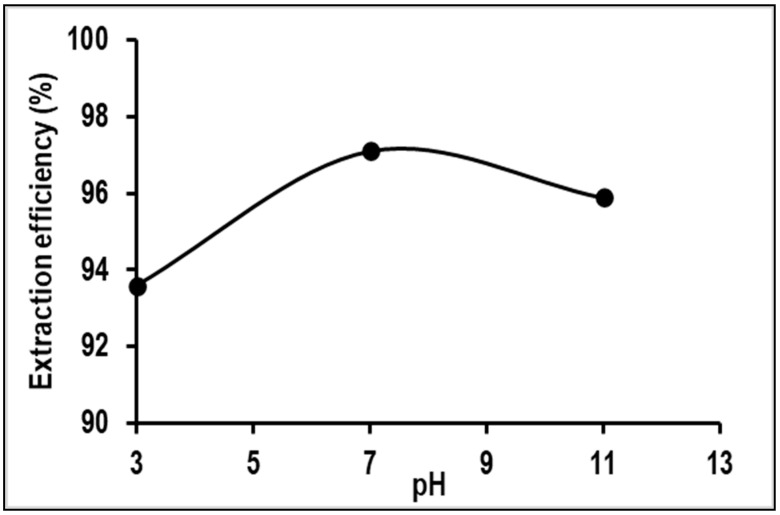
Effect of pH on the removal of Mn using PVA–alginate–[HIMP][TS].

**Figure 8 molecules-28-05777-f008:**
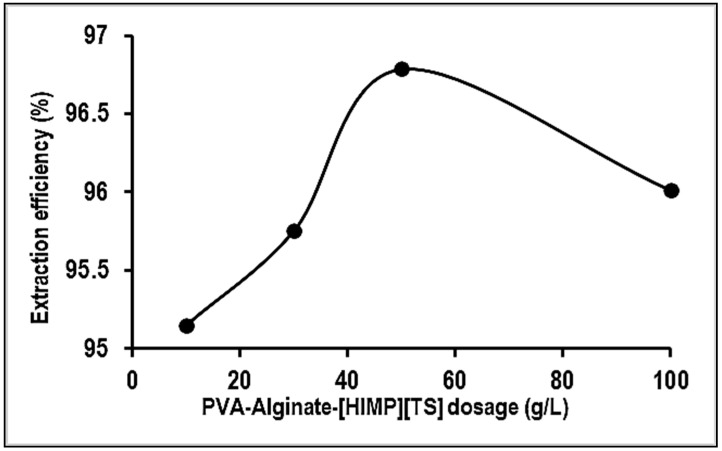
Effect of adsorbent dosage on the removal of Mn using PVA–alginate–[HIMP][TS].

**Figure 9 molecules-28-05777-f009:**
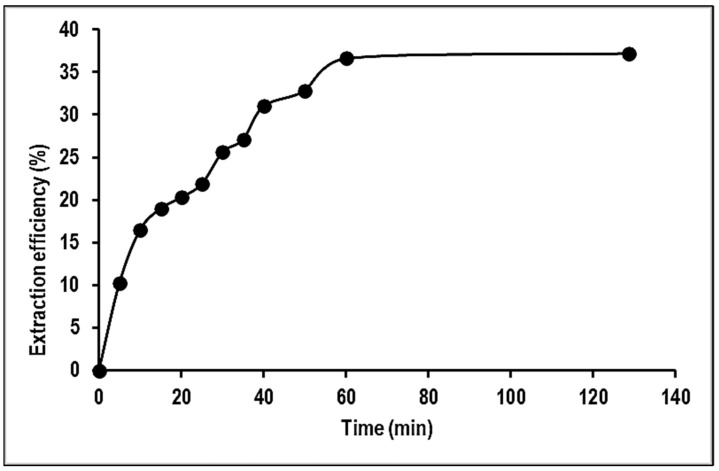
Effect of contact time on the removal of Mn using PVA–alginate–[HIMP][TS].

**Figure 10 molecules-28-05777-f010:**
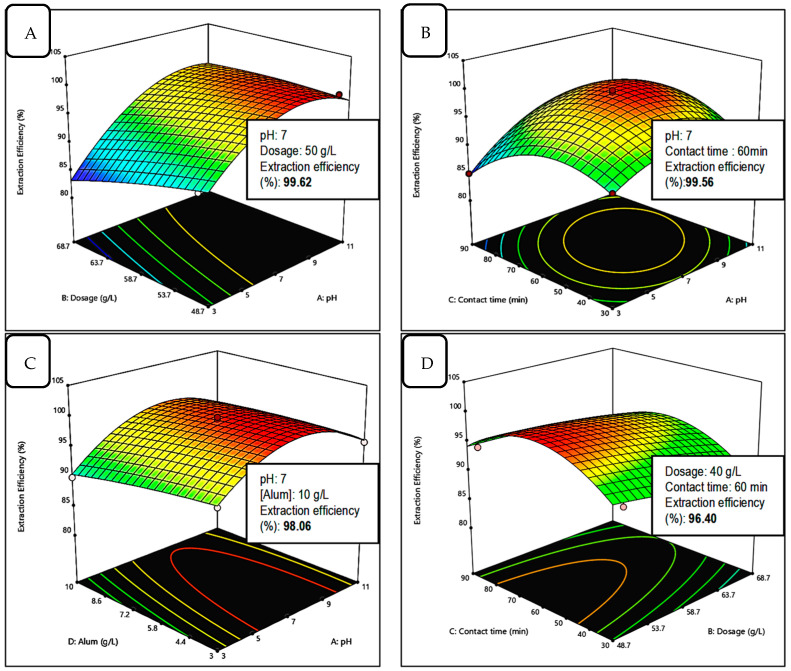
The 3D response surface plots of the interactions between significant independent variables (**A**) pH, (**B**) dosage, (**C**) contact time, and (**D**) aluminium concentration towards the dependent variable, and the extraction efficiency (%) of Mn from experimental design.

**Figure 11 molecules-28-05777-f011:**
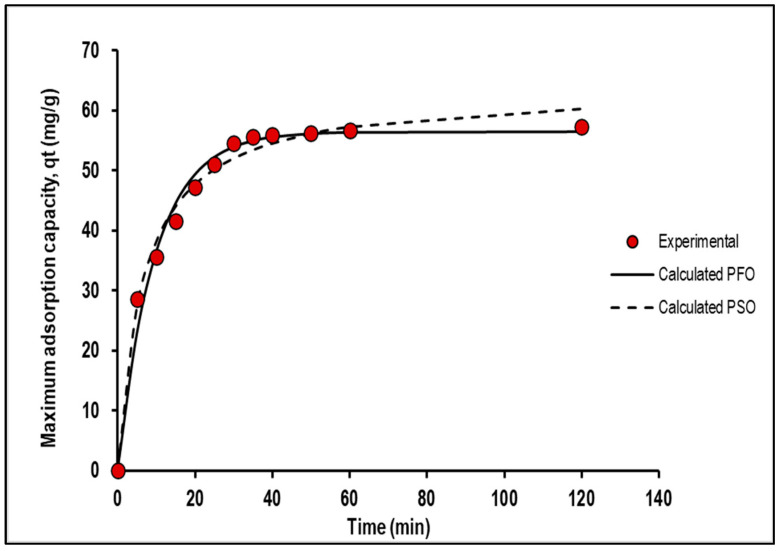
Experimental data versus calculated data on pseudo-first- and pseudo-second-order kinetic analyses of Mn adsorption study using PVA–alginate–[HIMP][TS].

**Figure 12 molecules-28-05777-f012:**
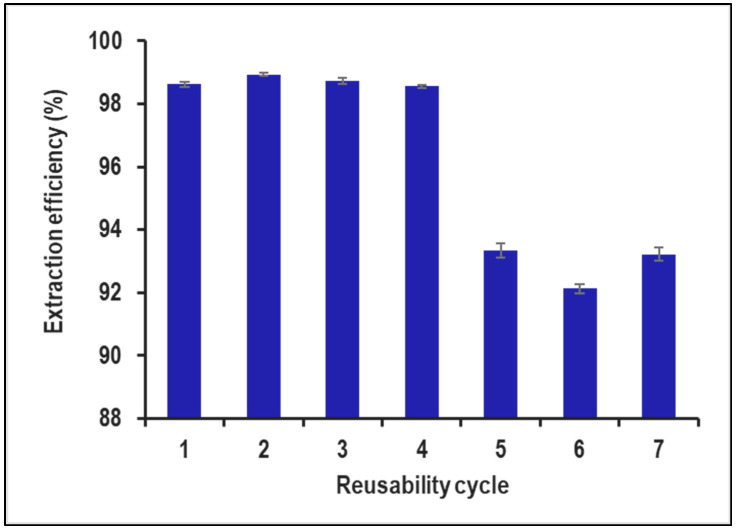
Reusability cycle study for PVA–alginate–[HIMP][TS].

**Figure 13 molecules-28-05777-f013:**
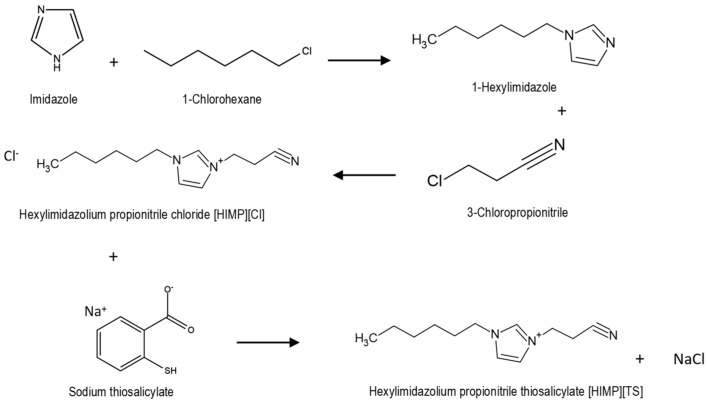
The reaction pathway for the synthesis of [HIMP][TS] ionic liquid.

**Figure 14 molecules-28-05777-f014:**
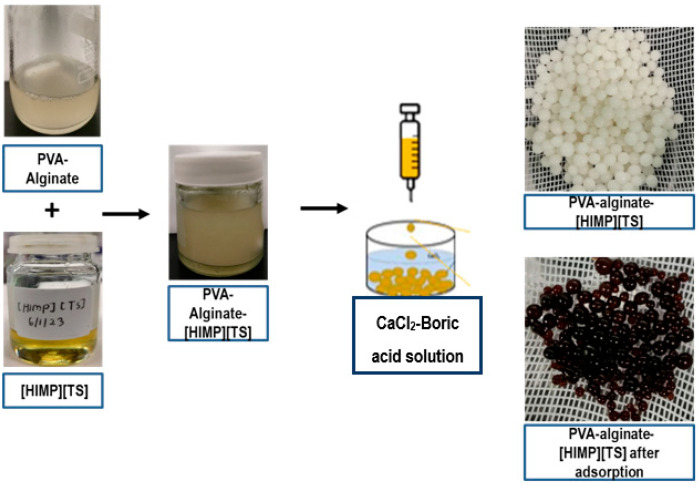
Schematic diagram of PVA–alginate–[HIMP][TS] beads.

**Figure 15 molecules-28-05777-f015:**
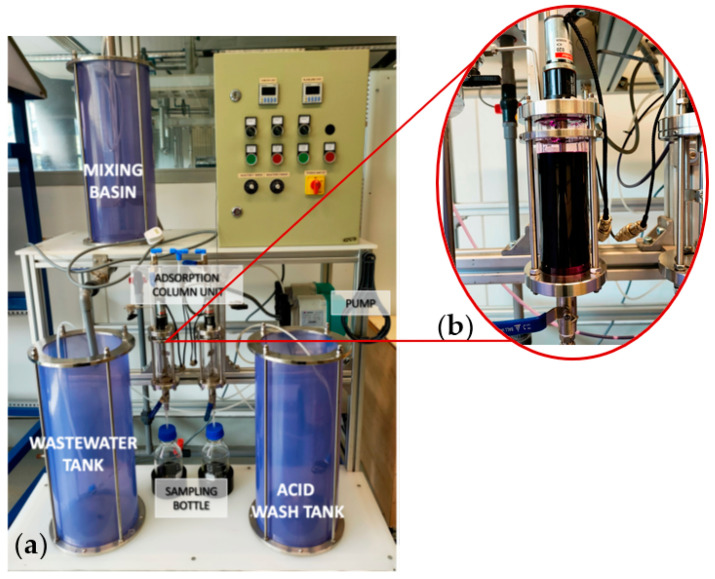
Wastewater filtration system scale-up prototype (**a**) and adsorption flask (250 mL) for the prototype (**b**).

**Table 1 molecules-28-05777-t001:** Experimental plan and results for the optimisation of Mn adsorption using Box–Behnken design.

Std	pH	Dosage (g/L)	Contact Time (min)	(Alum)	Actual (%)	Predicted (%)
1	3	30	60	6.5	94.53	94.15
2	11	30	60	6.5	95.1	93.93
3	3	50	60	6.5	89.87	90.22
4	11	50	60	6.5	98.06	97.63
5	7	40	30	3	95.4	95.02
6	7	40	90	3	94.99	94.70
7	7	40	30	10	93.69	93.16
8	7	40	90	10	92.86	92.42
9	3	40	60	3	93.84	94.14
10	11	40	60	3	95.78	96.12
11	3	40	60	10	89.91	90.46
12	11	40	60	10	95.08	95.67
13	7	30	30	6.5	94.72	94.90
14	7	50	30	6.5	92.34	93.06
15	7	30	90	6.5	92.48	92.64
16	7	50	90	6.5	93.54	94.25
17	3	40	30	6.5	90.93	90.56
18	11	40	30	6.5	88.29	88.67
19	3	40	90	6.5	84.99	84.54
20	11	40	90	6.5	93.32	93.62
21	7	30	60	3	99.81	100.47
22	7	50	60	3	99.15	98.52
23	7	30	60	10	96.01	96.57
24	7	50	60	10	99.01	98.28
25	7	40	60	6.5	99.56	99.39
26	7	40	60	6.5	99.43	99.39
27	7	40	60	6.5	99.89	99.39
28	7	40	60	6.5	99.31	99.39
29	7	40	60	6.5	98.78	99.39

**Table 2 molecules-28-05777-t002:** ANOVA result of the quadratic model of Mn removal percentage by PVA–alginate–[HIMP][TS] beads.

**Parameters**	Initial concentration
	Adsorbent dosage
	Contact time
	Alum concentration
**Model *F*-value**	54.05 (significant)
** *R* ** **-squared**	0.9819
**Adjusted *R*-squared**	0.9636
**Predicted *R*-squared**	0.9022
**CV**	0.76
**PRESS**	39.55
**Adequate precision**	30.56
**Lack of fit**	0.0946 (not significant)

**Table 3 molecules-28-05777-t003:** Comparison of optimum conditions and results obtained between OFAT and RSM for Mn removal.

Factors	OFAT	RSM
pH	7	7.15
Contact time	60 min	62.21 min
Dosage	50 g/L	38.26 g/L
Alum	5 g/L	4.54 g/L
Extraction percentage (%)	95.15	98.91

**Table 4 molecules-28-05777-t004:** Calculated maximum adsorption capacity, q_t_, and error function analysis of kinetic model of Mn adsorption using PVA–alginate–[HIMP][TS].

Model	Pseudo-First-Order	Pseudo-Second-Order
Parameter	2	2
Maximum adsorption capacity, q_t_ (mg/g)	54.41	63.61
RMSE	2.3	1.9
R^2^	0.98	0.99
Adjusted R^2^	0.98	0.99
AICc	30.39	20.61
HQC	21.03	17.25

**Table 5 molecules-28-05777-t005:** The performance of Mn removal by PVA–alginate–[HIMP][TS] beads on filtration system prototype with optimised condition parameters.

Parameter	Condition
Alum concentration	5 g/L
pH	7.15
Bead dosage	35 g/L
Contact time	120 min
**Performance**	
Extraction efficiency (%)	99.14%
Turbidity (NTU) (before; after)	(907; 0.13)
Total suspended solid (TSS)	(1844; 0)

**Table 6 molecules-28-05777-t006:** The coded level of independent variables of the Box–Behnken experimental design for Mn adsorption.

Independent Variables	Factor	Coded Levels		
		−1	0	+1
pH	A	3	7	11
Adsorbent dosage (g/L)	B	30	40	50
Contact time (h)	C	30	60	90
Alum (g/L)	D	3	6.5	10

## Data Availability

Not applicable.

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
