# Peer review of "Manganese Removal Using Functionalised Thiosalicylate-Based Ionic Liquid: Water Filtration System Application"

_molecules, 2023, doi:10.3390/molecules28155777_

Round 1

Reviewer 1 Report

Manuscript molecules-2449769 entitled “Manganese removal using functionalised thiosalicylate-based ionic liquid: Water filtration system application”.

Please notice the following:

General view: The manuscript illustrated a great idea to manganese removal from industrial wastewater, which is known toxic and can harm both the environment and human health. The topic is very interesting.This problem is relevant for journal scope. 

The Autors expressed their idea in moderate language and grammar. The manuscript might require copyediting and proofreading up to a little degree to provide more simplified sentences.

The introduction is easy detailed. The concept and aim are clearly defined.  The presentation and discussion of the presented topicis clear and very detailed.

Suggests supplementing the "Introduction" with information bringing a new scientific contribution. 

In general, the paper is well written, the results are conclusive and of interest for this topic. I have not found any important formal mistakes or typo errors. Formally, the paper is well written and easy to understand.

Please cite more papers from MDPI journals at the last 2-3 years in the similar topic of this research.

Other weaknesses to be corrected:

1. Keywords should be in alphabetical order.

2. Correct indexes in units

3. Please make a list of abbreviations at the end of the manuscript

The manuscript follows the formal regulations of MDPI journals.

I suggest the acceptance after minor revision

Reviewer 2 Report

The manuscript by Basirun et al. devoted to obtaining of new soebent based on sodium alginate, PVA and ionic liquid for Mn removal. The authors optimized sorption parametres and found the best parameters for metal ion uptaking. Moreover, they try to simulate industrial system for wastewater cleaning.

However, this work doesn't highlight influence of ionic liquid on sorption process. I recommend the authors compare sorbent with IL and without it in optimal conditions to show how the IL presence impact Mn sorption.

Also, in Fig. 12 typo in CaCl2

Round 2

Reviewer 2 Report

Despite the authors stated that information of Mn removal efficiency presented in the MS, I couldnt find it somewhere expecting Author response. 

Also, the authors said that this information is in the Introduction section. To my mind, it should be in the Results and Discussions section to highlight efficiency of the proposed sorbent and justify the use of expensive ionic liquid
